# Long-Time Dynamics of Selected Molecular-Motor Components Using a Physics-Based Coarse-Grained Approach

**DOI:** 10.3390/biom13060941

**Published:** 2023-06-05

**Authors:** Adam Liwo, Maciej Pyrka, Cezary Czaplewski, Xubiao Peng, Antti J. Niemi

**Affiliations:** 1Faculty of Chemistry, University of Gdańsk, Fahrenheit Union of Universities, Wita Stwosza 63, 80-308 Gdańsk, Poland; maciej.pyrka@ug.edu.pl (M.P.); cezary.czaplewski@ug.edu.pl (C.C.); 2Department of Physics and Biophysics, University of Warmia and Mazury, ul. Oczapowskiego 4, 10-719 Olsztyn, Poland; 3Center for Quantum Technology Research, Key Laboratory of Advanced Optoelectronic Quantum Architecture and Measurements (MOE), School of Physics, Beijing Institute of Technology, Beijing 100081, China; xubiaopeng@bit.edu.cn; 4Nordita, Stockholm University and Uppsala University, Roslagstullsbacken 23, SE-106 91 Stockholm, Sweden; antti.niemi@su.se

**Keywords:** molecular motors, falling cat motion, molecular dynamics, coarse graining, UNRES force field

## Abstract

Molecular motors are essential for the movement and transportation of macromolecules in living organisms. Among them, rotatory motors are particularly efficient. In this study, we investigated the long-term dynamics of the designed left-handed alpha/alpha toroid (PDB: 4YY2), the RBM2 flagellum protein ring from *Salmonella* (PDB: 6SD5), and the V-type Na${}^{+}$-ATPase rotor in *Enterococcus hirae* (PDB: 2BL2) using microcanonical and canonical molecular dynamics simulations with the coarse-grained UNRES force field, including a lipid-membrane model, on a millisecond laboratory time scale. Our results demonstrate that rotational motion can occur with zero total angular momentum in the microcanonical regime and that thermal motions can be converted into net rotation in the canonical regime, as previously observed in simulations of smaller cyclic molecules. For 6SD5 and 2BL2, net rotation (with a ratcheting pattern) occurring only about the pivot of the respective system was observed in canonical simulations. The extent and direction of the rotation depended on the initial conditions. This result suggests that rotatory molecular motors can convert thermal oscillations into net rotational motion. The energy from ATP hydrolysis is required probably to set the direction and extent of rotation. Our findings highlight the importance of molecular-motor structures in facilitating movement and transportation within living organisms.

## 1. Introduction

Molecular motors are nanomachines that convert chemical energy or light into motion. Natural molecular motors are usually protein or RNA assemblies [1,2,3,4]. Their roles include the transportation of intracellular substances, motion, mitosis, separating DNA strands, and the construction of the cytoskeleton. Molecular motors can be divided into linear and rotatory motors, depending on the kind of motion. Examples of the first class are kinesins [5] and dyneins [6] that transport cargo inside a cell using the microtubule tracks and myosin that governs muscle contraction [7]. The rotatory motors on which we focus here include cellular flagella [8,9], ATP synthases [10,11], and ATPases that act as ion transporters [12].

Molecular motors are powered by ATP hydrolysis [1,2,3,4]. However, the process of energy conversion differs from that of macroscale engines because of its stochastic nature [3,11,13,14]. Models of energy conversion have been developed, from which it follows that only a minimal energy input is required [11,13,14]. However, thus far, the detailed mechanisms of molecular-motor action has not been established.

In a previous work, it was found that ring structures can engage in rotational motion even with zero total angular momentum owing to the deformation of their shape [15,16,17,18]. This research was inspired partly by the seminal observation that rotational motion can emerge by shape deformations even in the absence of angular momentum made by Guichardet [19] and by Shapere and Wilczek [20], in combination with the idea of time crystals, whose existence was predicted by Shapere and Wilczek [21,22], according to which systems undergoing periodic rotational motion can constitute deep free-energy minima [20]. The reader is referred to the literature cited above and to a recent paper [23] for more information. These concepts were validated by carrying out microcanonical all-atom simulations of the cyclopropane molecule [17]. Subsequently, the cyclo-Ala${}_{9}$ and cyclo-Ala${}_{42}$ polypeptides with designed trefoil geometry were also found to exhibit net rotation in both microcanonical and canonical all-atom simulations in vacuo and with TIP3P water, respectively [18]. The canonical simulations demonstrated that the thermal motions due to the water molecules bumping onto the peptide, together with shape deformation, are converted into its net rotational motion. The results suggested that molecular rotatory motors can convert thermal motion into rotation owing to the fact that their shapes are deformable. It should also be noted that vortex states have recently been detected at low temperatures (85 K) by nuclear magnetic resonance (NMR) in all-hydrocarbon catenanes [24]. More recently, Courbet et al. [25] designed and synthesised model protein motors and demonstrated experimentally that they populate multiple rotational states that correspond to the rotation about the system axle.

The systems studied computationally in the papers mentioned above [15,16,17,18] were small. In this work, we make a step forward and study larger systems, more related to molecular rotatory motors, by using the coarse-grained UNRES model of polypeptide chains [26,27,28,29,30]. Owing to a high degree of coarse graining, with only two interaction sites per amino acid residue, UNRES enables us to run large-scale simulations in terms of system size and simulation time. Despite heavy coarse graining, UNRES reproduces protein structures very well [31] owing to the application of the scale-consistent theory in the UNRES force field [28,32], which enables us to embed atomistic details in the effective potential energy function in the form of non-radial and correlation terms. Moreover, owing to the elimination of fine-grained degrees of freedom from the coarse-grained model, one unit of UNRES simulation time amounts to about 1000 units of all-atom or laboratory time [33,34].

In this work, we studied the following three systems: the designed left-handed alpha/alpha toroid (PDB: 4YY2) [35], the RBM2 inner ring of the flagellum MS-ring protein FliF from *Salmonella* (PDB: 6SD5) [36], and the rotor of the V-Type Na${}^{+}$-ATPase from *Enterococcus hirae* (PDB: 2BL2) [12]. The first system is not a natural molecular rotating motor but is it similar to the smaller systems studied in a previous work [17]. The purpose of its selection was to compare the dynamics of this protein with that found for the cyclic alanine polypeptides in previous all-atom simulations. The two other systems are parts of real molecular rotatory motors [18]. We demonstrate that the first system exhibits a significant net rotational motion with zero total angular momentum in very long microcanonical simulations, while both 4YY2 and the two motors exhibit significant net rotational motion independent of their angular momentum in canonical Langevin simulations in which a mean-field lipid model was included in the case of the two motors [37].

## 2. Materials and Methods

### 2.1. Systems Studied

The experimental structures of the systems studied are shown in Figure 1A–C. The first system (PDB: 4YY2, 99 amino acid residues) is a designed 6-α-helix bundle with a toroid geometry. It has a triangular shape when viewed in the plane perpendicular to helices I, III, and V, which form an inner three-helix bundle, while helices II, IV, and VI are packed at an angle to helices I, III, and V. In its crystal structure, the outer helices are packed against those of the neighboring protein molecules. In our simulations, we considered only one monomer (chain A of the 4YY2 structure). The roughly triangular shape of a monomer makes it similar to a model triangle or to the cyclopropane molecule studied in an earlier work [15,16,17,18]. The protein is acyclic, so we imposed a weak N–C-terminus distance restraint in simulations to make it a ring (see Section 2.4). This system is hereafter referred to as 4YY2.

The experimental structure of the second system (the RBM2 inner ring of the flagellum MS-ring protein FliF from *Salmonella*; PDB:6SD5) [36] is a small part of a flagellum [8,9] and works as a rotor that transmits the torque to the flagellum shaft that is locked inside the ring. The structure is immersed in the inner cell lipid membrane. It consists of 22 monomers, each containing 98 amino acid residues, resulting in 2156 residues in total, with a structure composed of 2 α-helices packed against a four-stranded antiparallel β-sheet (Figure 1B). The monomers are arranged into a 22-membered symmetric ring, with an inner diameter of about 70 Å and an outer diameter of about 159 Å; the long axes of the monomers are in the ring plane (Figure 1B). Because the shaft was not present in our simulations, we had to introduce distance restraints to prevent the collapse of the ring (see Section 2.4). The system is hereafter referred to as 6SD5.

The experimental structure of the third system, the rotor of the V-Type Na${}^{+}$-ATPase from *Enterococcus hirae* (PDB: 2BL2) [12], is a barrel composed of 10 left-handed four-helix-bundle monomers, each containing 156 amino acid residues, resulting in 1560 residues in total. Helices II and IV form the outer wall, and helices I and III form the inner wall of the barrel. The inner and outer diameters are about 50 Å and 80 Å, respectively. The height of the barrel is about 60 Å (Figure 1C). The barrel pierces the lipid membrane, and a protein-complex shaft is inserted inside it. The system acts as a sodium-ion circular transporter, with one ATP-hydrolysis cycle corresponding to an anticlockwise rotation by 120${}^{\circ }$. This process moves a sodium ion to the delivery site [12]. Because the shaft was not included in our simulation, its presence was accounted for by introducing the respective distance restraints (see Section 2.4). The system is hereafter referred to as 2BL2.

### 2.2. UNRES Model of Polypeptide Chains

The UNRES model of polypeptide chains developed in our laboratory [26,27,28,29,30] uses a highly reduced representation of polypeptide chains, in which a chain consists of α-carbon (C${}^{\alpha }$) atoms linked by virtual bonds. The C${}^{\alpha }$s serve only to define the backbone geometry, while the interaction sites are the peptide groups (p) located halfway between the consecutive C${}^{\alpha }$s and the united side chains attached to the C${}^{\alpha }$s with the C${}^{\alpha }\cdots$SC virtual bonds (Figure 2). The sites and the corresponding interaction potentials have axial and not spherical symmetry.

The effective energy function of the UNRES model has been derived systematically as a truncated Kubo cluster cumulant expansion [38] of the potential of the mean force of a polypeptide in an appropriate environment (water or lipid membrane) [28,32,39]. Consequently, it represents free and not potential energy and depends on temperature [40]. The present version of UNRES contains the energy terms derived based on our recently developed scale-consistent theory of coarse graining [28,32], in which the atomistic details are rigorously embedded in the effective interaction potentials. The energy function consists of the site–site (SC–SC, SC–p, and p–p) interaction terms, the local terms corresponding to the energetics of virtual bonds, backbone virtual-bond angles, backbone virtual-bond dihedral angles, and side-chain rotamers, and of the multibody or correlation terms that couple the backbone–local and backbone–electrostatic interactions [28,39]. The correlation terms are necessary for the correct modeling of regular secondary structures [28,39,41]. The solvent is implicit in the interaction potentials, and the present parameterization corresponds to physiological pH. The details of the UNRES model and force field are available in the references cited [27,28,29,30].

**Figure 2 biomolecules-13-00941-f002:**
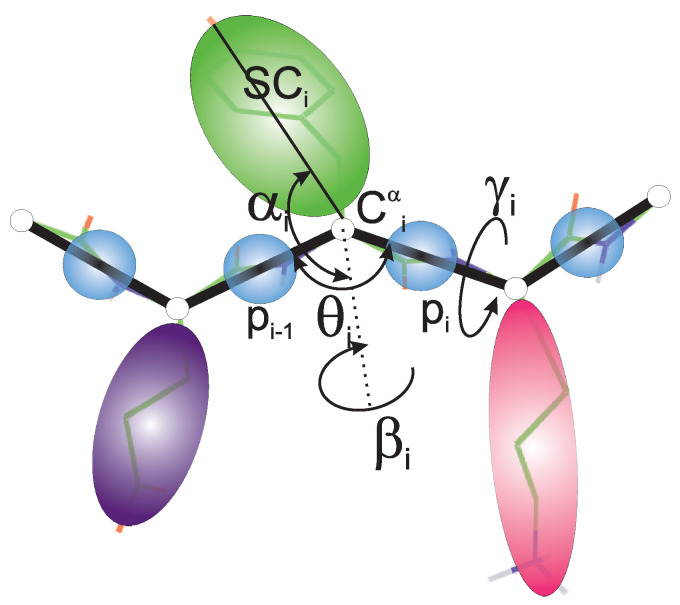
UNRES model of polypeptide chains. The interaction sites are united peptide groups located between the consecutive α−carbon atoms (light-blue spheres) and united side chains attached to the α−carbon atoms (spheroids with different colors and dimensions). The backbone geometry of the simplified polypeptide chain is defined by the C${}^{\alpha }\cdots$C${}^{\alpha }\cdots$C${}^{\alpha }$ virtual-bond angles θ ($\theta_{i}$ has its vertex at C${}_{i}^{\alpha }$) and the C${}^{\alpha }\cdots$C${}^{\alpha }\cdots$C${}^{\alpha }\cdots$C${}^{\alpha }$ virtual-bond-dihedral angles γ ($\gamma_{i}$ has its axis passing through C${}_{i}^{\alpha }$ and C${}_{i+1}^{\alpha }$). The local geometry of the *i*th side-chain center is defined by the polar angle $\alpha_{i}$ (the angle between the bisector of the angle $\theta_{i}$ and the C${}_{i}^{\alpha }\cdots$SC${}_{i}$ vector) and the azimuth angle $\beta_{i}$ (the angle of counter-clockwise rotation of the C${}_{i}^{\alpha }\cdots$SC${}_{i}$ vector about the bisector from the C${}_{i-1}^{\alpha }\cdots$C${}_{i}^{\alpha }\cdots$C${}_{i+1}^{\alpha }$ plane, starting from C${}_{i-1}^{\alpha }$). For illustration, the bonds of the all-atom chains, except for those to the hydrogen atoms connected with the carbon atoms, are superposed on the coarse-grained picture. Reprinted with permission from Ref. [42]. Copyright 2015 American chemical Society

In this work, we used the latest NEWCT-9P variant of the UNRES force field, which has been parameterized by the maximum likelihood method [42] with 9 proteins of all secondary-structure types [29]. The calibration was aimed at reproducing the experimental ensembles of the training proteins determined by nuclear magnetic resonance (NMR). The ensembles of three of the training proteins were determined at a range of temperatures encompassing those below the melting temperature, the melting temperature, and those above the melting temperatures. Thus, the force field is able to model not only the folded state but also the folding process and major conformational changes.

### 2.3. Coarse-Grained Molecular Dynamics with UNRES

Because the solvent is implicit in UNRES, Langevin dynamics with the UNRES energy function as the potential energy is the obvious way to simulate motion with this model. This approach is justified on theoretical grounds because the forces acting on a coarse-grained system can be separated into the mean forces (the spatial derivatives of the potential of mean force, which is approximated by the UNRES effective energy function) and those coming from the memory kernel. With an infinitely narrow memory window, the latter are reduced to stochastic and friction forces [43,44,45,46,47]. Molecular dynamics with UNRES was developed in our earlier work [33,48], and the equations of motion as well as the algorithms for their numerical solution have been revised and optimized for performance and memory recently [34]. The variables are the Cartesian coordinates of the C${}^{\alpha }$ atoms and those of the SC centers. With these variables, the inertia matrix is a symmetric five-band matrix. This feature results from the fact that the peptide-group centers are off the C${}^{\alpha }$ atoms in the UNRES model and that all virtual bonds are modeled by stretchable rods with masses distributed along the rods [48]. Distributing the mass of a site instead of locating it in the site’s center provides a description of motion closer to that of the all-atom representation, which is important in view of the observation from an earlier work [15,16,17,18] that local deforming motions compensate for the rotations of the whole system to keep zero total angular momentum.

The Langevin equations of motion of a multichain protein in the UNRES model are given by Equation (Equation 1). It should be noted that, because different chains are not bonded, the equations of motion of different chains are not coupled through the inertia matrix but only through the potential forces. Likewise, the equations of motion for the *x*, *y*, and *z* coordinates are not coupled through the inertia matrix but only through the potential forces.
(1)$$G_{I}{\ddot{q}}_{I\xi }=-\nabla_{q_{I\xi }}U-A_{I}^{T}\Gamma_{I}A{\dot{q}}_{I\xi }+A_{I}^{T}F_{I\xi }^{r},\;I=1,2,\ldots \;N_{c};\;\xi =x,y,z$$
where $\dot{q}$ and $\ddot{q}$ denote the velocities and accelerations of the generalized coordinates (which are the Cartesian coordinates of the C${}^{\alpha }$ atoms and those of side-chain centers), $A_{I}$ is a constant matrix that relates the generalized coordinates of chain *I* to site Cartesian coordinates, $G_{I}$ is the inertia matrix for chain *I*, $\Gamma_{I}$ is the diagonal matrix of site friction coefficients for chain *I*, *U* is the UNRES potential energy function, $F_{I}^{r}$ are the random forces acting on the sites of chain *I*, and $N_{c}$ is the number of chains. Details of the approach can be found in our recent paper [34]. In microcanonical mode, the friction and stochastic forces do not appear on the right-hand side of Equation (Equation 1).

### 2.4. Simulation Procedure

#### 2.4.1. The 4YY2 System

For the designed 4YY2 protein, we first assessed the ability of UNRES to model its tertiary structure in order to determine whether the actual MD simulations could be run in unrestrained mode. To do this, we applied the UNRES-based protein structure prediction protocol developed in our earlier work [49]. Briefly, the protocol consists of multiplexed replica-exchange molecular dynamics (MREMD) simulations as implemented with UNRES [50] followed by post-processing the results with the weighted histogram analysis method (WHAM) [51] and Ward minimum variance clustering [52] to identify five prediction candidates. We ran 12 temperature replicas, ranging from 260 to 370 K, each multiplexed 4 times (for a total of 48 trajectories). The temperatures were distributed according to the Hansmann algorithm to visit all temperatures by every trajectory [53]. Each trajectory consisted of 40,000,000 steps of 4.89 fs length. Every trajectory was started from a different randomly generated structure, and no information about the experimental structure of the protein was included. The top prediction (which corresponded to the cluster with the lowest free energy at T=280 and 300 K) had a C${}^{\alpha }$ root-mean-square deviation (RMSD) of 6.7 Å and a global distance test total score (GDT_TS) [54] of 40.4, which is a good result for a 99-residue protein (with the values corresponding to a comparison with chain A of the experimental structure). This structure is superposed on the experimental 4YY2 structure (chain A) in Figure 3. As can be seen from the Figure, the predicted structure has the correct secondary structure and correct arrangement of the six helices. Out of 87 residues of chain A of the experimental 4YY2 structure, for which the DSSP algorithm [55,56] assigns a helix-related secondary structure (an α-helix, 3–10 helix, turn, or hydrogen-bonded turn), 88.5% of the model has a helix-related structure. The only difference between the tertiary structure of the experimental and predicted structures is that helices II, IV, and VI in the predicted structure are not at an angle to helices I, III, and VI in the predicted structure. It should be noted that the change in the packing of helices II, IV, and VI could very well be caused by the fact that helices II, IV, and VI are tightly packed against those of the neighboring molecules in the crystal structure, while only one molecule was considered in our MREMD simulations.

To run actual molecular dynamics simulations, we started from chain A of the experimental structure, on which we imposed a flat-bottom distance restraint [60] with a well depth of 50 kcal/(mol×Å${}^{2}$) and lower and upper distance boundaries of 3.8 and 7.4 Å, respectively. After carrying out energy minimization, we ran a short (10,000 step) canonical MD simulation at T=100 K to obtain a stable temperature. Subsequently, we zeroed the mass-center velocity and the total angular momentum of the system and started microcanonical (NVE) trajectories, in which the effective temperature was always close to 100 K even though the temperature was no longer controlled. Both the velocity of the mass center and the total angular momentum were effectively zero, and the total energy only oscillated about the shadow-Hamiltonian value with an amplitude of about 0.002 kcal/mol within a run, with no drift being observed. However, the calculation had to be split due to the wall-clock-limit constraint imposed by the batch-queue system. Consequently, it had to be restarted after every 400,000,000 steps. The round-off errors arising from dumping the coordinates and velocities for restart with a reduced precision resulted in a residual angular moment after the restart information had been read, which had to be zeroed out by adjusting the velocities. As a result, the total energy after the restart could differ by up to 0.05 kcal/mol from that of the previous section of a run. We ran 6 trajectories (both in the initial thermalization and in the production phase), each lasting 4,000,000,000 steps of 1 fs length; we also kept the 1 fs time step in all other MD simulations (both microcanonical and canonical). Even though a time step up to 10 fs can be applied with UNRES, we chose to apply a small time step to avoid possible problems with the conservation of energy, velocity of the mass center, and angular momentum. The total simulation time amounted to 4 μs. However, because of the dilatation of the UNRES time scale [34,48], this time corresponds to about 4 ms of laboratory time scale. Another series of 6 runs was carried out in canonical Langevin-dynamics mode at T=300 K, scaling the water-friction coefficient by a factor of 0.02. The duration of this run was 0.04 μs, and the calculation was run in a single shot, so no restart was necessary.

#### 2.4.2. The 6SD5 System

Because the system is large and is only a part of the bacterial flagellum, we did not attempt at assembling it from scratch but took the experimental 6SD5 structure and imposed restraints to account for the protein shaft present inside the ring [36] and to keep the ring components together. We imposed two kinds of restraints. The first set consisted of weak harmonic restraints on the pairs of monomers facing one another across the ring, in which all C${}^{\alpha }\cdots$C${}^{\alpha }$ distances smaller than 75 Å were restrained at the experimental-structure values with a force constant of 0.1 kcal/(mol×Å${}^{2}$). These restraints are hereafter referred to as type I restraints. Restraints of the second type were imposed on the distances between the C${}^{\alpha }$ atoms from the ring pivot (the normal to the ring plane passing through the ring center). These restraints are hereafter referred to as cylindrical restraints or type II restraints. For the 6SD5 system, harmonic restraints with a force constant of 0.1 kcal/(mol×Å${}^{2}$) were additionally imposed on the distances between the C${}^{\alpha }$ atoms of the neighboring monomers. The cylindrical restraints are defined by Equation (Equation 2).
(2)$$V_{cyl}=\frac{1}{2}k\sum_{i}{\left(\sqrt{x_{i}^{2}+y_{i}^{2}}-\sqrt{{\left(x_{i}^{\circ }\right)}^{2}+{\left(y_{i}^{\circ }\right)}^{2}}\right)}^{2}$$
where $x_{i}$ and $y_{i}$ are the respective *x* and *y* coordinates of the *i*th C${}^{\alpha }$ atom in the coordinate system with its origin at the ring center and with the ring pivot as the *z* axis; $x_{i}^{\circ }$ and $y_{i}^{\circ }$ are the respective *x* and *y* coordinates of the initial (experimental) structure, and *k* is the force constant. We set k=1 kcal/(mol×Å${}^{2}$). Four trajectories were run in the microcanonical mode with type I restraints at an effective temperature of T=300 K. As for the 4YY2 system, the total energy was conserved, and the oscillations had an amplitude of about 0.05 kcal/mol due to the bigger system size. The initial thermalization (prior to the production microcanonical runs) was carried out as for the 4YY2 system. Two more series of runs were carried out in the NVT regime using the UNRES model of the lipid membrane, with a non-polar phase thickness of 30 Å a lipid-head thickness of 5 Å, and a scale factor of the lipid screening of peptide-group interactions of 0.7, as recommended in our earlier work [37]. One series of 4 trajectories was carried out with type I distance restraints, and the other series was carried out with type II (cylindrical) restraints. The duration of the microcanonical and canonical runs was 0.18 and 0.10 μs (amounting to 0.18 and 0.10 ms of laboratory time, respectively), with a time step of 1 fs. The initial structure was prepared with the CHARMM-GUI system [61], which was recently enhanced with input to UNRES [62]. For each series of calculation, the trajectory with a given index was started from the same geometry and velocities.

#### 2.4.3. The 2BL2 System

Similarly as for 6SD5, we carried out a series of 4 microcanonical runs with weak type I distance restraints with an effective temperature of T=300 K and two series of four canonical simulations at T=300 K in the lipid membrane, with one with type I and the other one with type II (cylindrical) restraints. The total energy was conserved in the microcanonical simulations, with the oscillations having an amplitude of about 0.05 kcal/mol. The cut-off distance below which the distances between the C${}^{\alpha }$ atoms of the monomers facing each other across the rings were restrained with type I restraints was 56 Å. Because the system was more self-contained, we did not impose distance restraints between the neighboring monomers in the run series along with type II restraints. The duration of microcanonical and canonical simulations was 0.18 μs (0.18 ms of laboratory time). The initial structure was prepared with the CHARMM-GUI system [61], which was recently enhanced with input to UNRES [62]. For each series of calculation, the trajectory with a given index was started from the same geometry and velocities.

### 2.5. Hardware Platform and Timing

All calculations were carried out with the Tryton Linux cluster (Megatel/Action, Cracow, Poland) located in the Academic Computer Center in Gdańsk, TASK. Each 6-trajectory 4 μs run for the 4YY2 system took 304 wall-clock hours with two Intel${}^{®}$ Xeon${}^{®}$ E5 v3 @ 2.3 GHz, 24-core (Haswell) processors. A 4-trajectory 6SD5 and 2BL2 run took 240 and 120 wall-clock hours per 0.1 μs with 24 cores/trajectory. Due to the long duration of simulations and queuing system limitations, the microcanonical simulations for the 4YY2 system had to be restarted after every 500,000,000 MD steps, and those for 6SD5 and 2BL2 had to be restarted after every 10,000,000 MD steps.

### 2.6. Determining the Rotation Angle

To determine the total rotation angle of the 4YY2 system, in which the plane of the ring changed its orientation considerably, we used the procedure developed in [18]. First, we selected three anchor points to define a triangle. The points were arranged in anticlockwise order, which enabled us to define a right-handed coordinate system centered in the geometric center of the triangle, with the *y* axis pointing from the center to point 1 (vector $r_{1}$), the *z* axis pointing along the vector product of the vectors $r_{1}$ and $r_{2}$, where vector $r_{2}$ runs from the center to point 2, and the *x* axis pointing along the vector product of the axes *y* and *z*. The anchor points of the 4YY2 system were the centers of helices II, IV, and VI calculated over the positions of the C${}^{\alpha }$ atoms in the order indicated. The triangle and the axes of the coordinate systems are superposed on the first frame of the MD trajectory shown in Figure 5. The coordinate system corresponding to the first frame was saved as the reference coordinate system. For each of the subsequent frames, the quaternion (Equation (Equation 3)) was determined from the reference and the current coordinate systems (with index *i*) using the Shepperd algorithm [63].
(3)$$\begin{array}{c} q_{i}=q_{\circ i}+q_{xi}i+q_{yi}j+q_{zi}k \end{array}$$
(4)$$\begin{array}{c} q_{\circ i}=\cos\frac{\vartheta }{2},\;q_{\xi i}=\sin\frac{\vartheta }{2}Q_{\xi i},\;\xi =x,y,z,\;Q_{x}^{2}+Q_{y}^{2}+Q_{z}^{2}=1 \end{array}$$
(5)$$\begin{array}{c} ii=jj=kk=-1,\;ij=-ji=k,\;jk=-kj=i,\;ki=-ik=j \end{array}$$

The real part of the quaternion ($q_{\circ i}$) is the cosine of half of the total rotation angle (ϑ), while the imaginary part defines the Euler rotation axis ($Q_{x},Q_{y},Q_{z}$) of the system from the reference frame. Thus, the quaternion provides an elegant description of the extent of rotation in terms of a single angle [64]. Because the quaternion has a two-fold degeneracy so that (ϑ,Q)≡(−ϑ,−Q), we required that the angle between the Euler axes of the two consecutive frames be less than $90^{\circ }$. When the rotation axis does not change to any greater extent except for small fluctuations (as we observed for the 6SD5 and 2BL2 systems), ϑ is the angle of the rotation about this axis.

For the 6SD5 and 2BL2 systems, rotation effectively occurred about the pivot to the ring. Both systems are composed of units that are bound by non-covalent interactions and, therefore, different monomers can exhibit different extents of rotation. For these reasons, we computed the rotation angle as given by Equation (Equation 6).
(6)$$\begin{array}{ccc} \vartheta_{m} & = & \frac{1}{N}\sum_{i=1}^{N}\theta_{mi} \end{array}$$
(7)$$\begin{array}{ccc} \theta_{mi} & = & \left\{\begin{array}{cc} 0 & m=1\\ \sum_{j=2}^{m}\left(\varphi_{ji}-\varphi_{j,i-1}+k_{ji}360^{\circ }\right) & m>1 \end{array}\right. \end{array}$$
(8)$$\begin{array}{ccc} \varphi_{ji} & = & \mathrm{atan}2\left(y_{ji},x_{ji}\right) \end{array}$$
where $\vartheta_{m}$ is the average rotation angle about the ring pivot at snapshot *m*, *i* is the monomer index, *N* is the number of monomers (10 for 2BL2 and 22 for 6SD5), $\theta_{mi}$ is the total angle of the rotation of the center of the *i*th monomer about the pivot from its initial position, $k_{ji}$ is an integer selected so that the absolute value of the difference between the rotation angles in two consecutive snapshots does not exceed $180^{\circ }$, $\varphi_{ji}$ is the angle of anticlockwise rotation of the center of monomer *i* at the *j*th snapshot about the *z* axis, $x_{ji}$ and $y_{ji}$ are the coordinates of the center of the *i*th monomer at the *j*th snapshots in the right-handed coordinate system of the first frame, with the *z* axis running along the ring pivot and origin in the current ring center, and the function atan2 is defined by Equation (Equation 9).
(9)$$\mathrm{atan}2\left(y,x\right)=\left\{\begin{array}{cc} \arctan\left(\frac{y}{x}\right) & \mathrm{if}\;x>0\\ \arctan\left(\frac{y}{x}\right)+\pi & \mathrm{if}\;x<0\;\mathrm{and}\;y\geq 0\\ \arctan\left(\frac{y}{x}\right)-\pi & \mathrm{if}\;x<0\;\mathrm{and}\;y<0\\ \frac{\pi }{2} & \mathrm{if}\;x=0\;\mathrm{and}\;y>0\\ -\frac{\pi }{2} & \mathrm{if}\;x=0\;\mathrm{and}\;y<0\\ \mathrm{undefined} & \mathrm{if}\;x=0\;\mathrm{and}\;y=0 \end{array}\right.$$

Thus, the sum over *j* (snapshots) defines the total angular travel of the center of a given monomer in the xy plane relative to the origin, and ϑ is the average over all monomers. To find how the rotation angles vary from monomer to monomer, we computed the standard deviation of θ for each snapshot, as given by Equation (Equation 10).
(10)$$\sigma_{\vartheta_{m}}=\sqrt{\frac{1}{N}\sum_{i=1}^{N}{\left(\theta_{mi}-\vartheta_{m}\right)}^{2}}$$

Subsequently, for a given simulation series, we computed the standard deviation averaged over all snapshots and all trajectories.

## 3. Results

### 3.1. The 4YY2 System

The plots of the C${}^{\alpha }$-RMSD from the experimental 4YY2:A structure for all six trajectories are shown in Appendix A. As shown, the RMSD does not increase beyond 4.3 Å, with the average over all trajectories being 2.8 Å. Thus, the protein’d conformation did not change remarkably during the simulations. In particular, its secondary structure and helix-packing topology, along with its pseudo-threefold symmetry, was conserved.

A plot of the total rotation angle ϑ about the Euler axis (Section 2.6) in simulation time for all six microcanonical trajectories is shown in Figure 4A. It can be seen that the rotation is significant for all trajectories, although, as opposed to the earlier work on alanine polypeptides with trefoil geometry [18], only less than half of a turn has been reached. The rotation is not only the rotation about the pseudo-threefold-symmetry axis (Figure 1A) but also involves the change in the orientation of the normal to the triangle plane, as shown in Figure 4B.

For illustration, selected structures from trajectory 4 are shown in Figure 5. As can be seen from the figure, the triangle defined by the centers of helices II, IV, and VI (see Figure 1A for helix numbering) rotates clockwise about its normal (the second snapshot at 1.5 μs); then, the plane of the triangle joins the rotation first about the *y* axis (the third snapshot at 2.5 μs) and then about the *y* and *x* axes (snapshot 4 at 3.0 μs). The rotation in the triangle plane accompanies those two, and when the plane is restored to the original orientation (snaphsot 5 at 3.4 μs), it reaches about $90^{\circ }$. Subsequently, the triangle plane rotates again (snapshot 6 at 4.0 μs). It should be noted that the total angular momentum of the protein is zero, and the observed apparent rotation is due solely to internal shape deformation.

The second series of simulations was carried out in the Langevin mode at T=300 K. Here, the total angular momentum was not controlled, and we were seeking to determine if the random thermal motions can be converted into net rotation. The variation of the RMSD from the experimental structure with time is shown in Appendix A. As shown, the RMSD is bigger than that for microcanonical simulations at an effective temperature of 100 K, with the average RMSD being 6.8 Å. Nevertheless, the secondary structure and the topology of helix packing are conserved. As in the case of the structure predicted using our UNRES-based protein structure prediction protocol [49] (Figure 3), the structures of the canonical simulations differ mainly by the packing of helices II, IV, and VI in a manner nearly parallel and not at an angle to helices I, III, and V. The variations of the total rotation angle ϑ and of the normal to the angle plane with the *z* axis are shown in Figure 6. As shown, net rotation occurs, and its extent is comparable to that obtained in microcanonical simulations. However, the duration of the canonical simulations was only 0.04 μs, as opposed to 4 μs for the microcanonical simulations. Similar differences between the time scale of net rotation of microcanonical and canonical simulations were observed in the recent work on cyclic alanine polypeptides [18].

**Figure 5 biomolecules-13-00941-f005:**
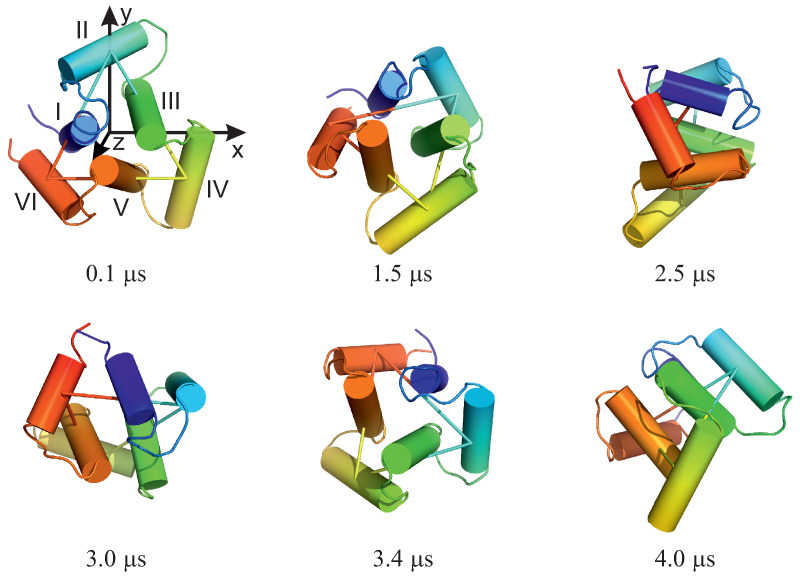
Selected snapshots of trajectory 4 of microcanonical simulations of the 4YY2 system. For convenience, the helices are labelled with the Roman numerals, and the axes of the coordinate system are indicated in the first snapshot. The drawings were made with PyMOL [57] after converting the coarse-grained structures to the all-atom representation using PULCHRA [58] and SCWRL [59].

It should be noted that the net rotation occurs in either direction, depending on the initial conditions (mainly initial velocity distribution), which depend on trajectory. Moreover, the rotation is not uniform and occurs back and forth, with this feature being more visible in canonical simulations, although its direction over a long time course does not change.

### 3.2. The 6SD5 System

The average monomer rotation angle about the normal to the ring plane in the entire 0.18 μs microcanonical simulation period was 3${}^{\circ }$, while its standard deviation was $6.3^{\circ }$, as illustrated in Appendix A. Therefore, even though a trend was observed that the average monomer rotation angle increased in absolute value, it could not be determined whether net rotation occurs in this mode. Judging from the duration of the microcanonical simulations of the 4YY2 system (Section 3.1), the trajectories should be run at least ten times longer (i.e., 1.8 μs) for the net rotation to exceed the standard deviation remarkably. On the other hand, in reality, the ring is sitting in the lipid membrane and interacts strongly with the other components of the flagellum, especially with the protein complex that constitutes the flagellar shaft. In this complex environment, it is not possible to keep zero total angular momentum. Therefore, we carried out the canonical simulations with distance (type I) and cylindrical (type II) restraints, as described in Section 2.4.2.

The plots of the variation of the C${}^{\alpha }$-RMSD from the initial structure are shown in Appendix A, respectively, while those of the variation of the C${}^{\alpha }$-RMSD of each of the 22 monomers from the initial structures are shown in Appendix A, respectively. It can be seen from Appendix A that the RMSD of the trajectories simulated with set I restraints (the distance restraints between the monomers facing each other across the ring; see Section 2.4.2) reached up to 13 Å. Given the large size of the ring (2156 residues total), this is not a big deviation, and neither monomer structure nor monomer-packing topology was affected. The plots of the monomer RMSDs (Appendix A) show that RMSD increases only up to 4 Å. It can also be noted that both the RMSD of the entire system and of the monomers initially increases and then mainly oscillates. The reason for the initial increase is the equilibration of the structure. The oscillations (although far from regular) of the complete structure of the ring and of the monomers, together with thermal motions, are probably responsible for the net rotation.

The RMSD of the structures of the second series with type II restraints (restraining the distances of the monomers from the ring pivot and the distances between the neighboring monomers) was below 1.5 Å due to the distance restraints imposed between neighboring monomers. Moreover, even without these additional restraints, the type II restraints were tighter than the type I restraints because they were imposed on the distances between the C${}^{\alpha }$-atoms and a fixed pivot. The RMSDs calculated over individual monomers (Appendix A) also were smaller than those corresponding to type I restraints (Appendix A). As stated in Section 2.4.2, without the additional restraints on neighboring monomers, the ring had a tendency to break when the type II restraints were imposed.

The plots of the monomer rotation angle relative to the initial position averaged over those of all monomers for the series of simulations with type I restraints and those with type II (cylindrical) restraints and distance restraints between the neighboring monomers are shown in Figure 7A,B, respectively.

It can be seen from Figure 7 that the rotation angle increased in absolute value in all simulations regardless of restraint type. The largest variation to about $-60^{\circ }$ occurred for trajectory 3 while the smallest occurred up to about $8^{\circ }$ at the end of the simulation for trajectories 1 and 2, respectively. However, even the smallest angle variation remarkably exceeded the standard deviation over the monomer, which was about $2^{\circ }$, on average, for the simulation series with type I restraints and $0.2^{\circ }$ for that with type II restraints. The direction and extent of the angle variation depended on the trajectory, which is understandable because each trajectory was started with a different set of random velocities. Remarkably, the corresponding trajectories from both simulations are very much alike, which strongly suggests that imposing the restraints had only a minor effect on the course of angle variation. Even with type I restraints, in which the ring pivot was not fixed, the rotation occurred virtually only about the ring pivot, and the pivot changed its orientation only by $4^{\circ }$ maximum, usually staying perfectly aligned with the original pivot. On the other hand, the center of ring did travel in the ring plane during the simulations with type I restraints.

It can be seen from Figure 7 that all graphs of angle variation with simulations exhibit quite regular ratcheting patterns (with a period of about 0.01 μs). These patterns seem to result from the fact that the random number generator was restarted with every run restart, which occurred after every 10,000,000 MD steps.

### 3.3. The 2BL2 System

As for the 6SD5 system, microcanonical simulations resulted in only small systematic changes in the average monomer rotation angle about the ring pivot, reaching up to $6^{\circ }$ at the end of the simulations, with the standard deviation being 6.8${}^{\circ }$. Thus, even though a trend was observed, much longer simulations would be required to achieve a remarkable extent of rotation about the pivot (Appendix A). Because the system pierces a lipid membrane and, consequently, neither the total energy nor the total angular momentum of the protein complex can be conserved, we carried out canonical simulations in the lipid-membrane model according to the procedure detailed in Section 2.4.3.

The plots of C${}^{\alpha }$-RMSD vs. simulation time for the canonical simulation series carried out with type I and type II restraints (see Section 2.4.2) are shown in Appendix A, respectively, while those for the individual monomers are shown in Appendix A, respectively. As shown, the RMSD did not exceed 9 Å for the type II and 14 Å for the type I restraints. The RMSD for the simulations with type II restraints is greater than that for the 6SD5 system because no restraints were imposed on the distances between the neighboring monomers. Both monomer structure and monomer-packing topology are preserved in all simulations. Similarly, the monomer RMSD in simulations with type II restraints (Appendix A) is smaller than in those with type I restraints (Appendix A).

The variation of the average monomer-rotation angle for the simulation series with type I and type II restraints is shown in Figure 8. It can be seen that the extent of rotation is greater compared to that of the 6SD5 system even during the first 0.1 μs of simulations. The rotation angle reached $140^{\circ }$ for trajectory 1 and $-120^{\circ }$ for trajectory 2, respectively, independent of restraint type. The standard deviations of the monomer rotation angle were $3.8^{\circ }$ for type I and $0.7^{\circ }$ for type II restraints, respectively, and thus clearly smaller than the extent of angle variation. As for the 6SD5 system, the trajectories of the rotation angle obtained with type I restraints are very similar to their counterparts obtained with type II restraints. The rotation occurred exclusively about the ring pivot, with the orientation of the pivot being virtually unchanged. Regular ratcheting patterns with a period of about 0.01 μs, different for every trajectory, are observed, which are probably due to restarting the random number generator with run restarts (every 10,000,000 time steps).

For trajectories 1 and 2, which exhibit the greatest extent of rotation, it can be seen that the long-time trend of angle change varies; for trajectory 2, the angle variation slows down after reaching about $-120^{\circ }$, while that for trajectory 1 accelerates to reach nearly linear variation after 0.1 μs. This feature, along with the different extent and direction of angle variation obtained for different trajectories, suggests that the energy input due to ATP hydrolysis is required only to direct the rotation and to provide the initial angular momentum, while the motor generally converts thermal motion into net rotation.

For illustration, three snapshots from trajectory 1 (with the greatest extent of ring rotation) obtained in the simulation with type II restraints are shown in Figure 9.

## 4. Discussion and Conclusions

In this study, using coarse-grained molecular dynamics with the UNRES model [26,27,28,29,30], we extended the size scale of earlier simulations of molecules and molecular systems with ring geometry [15,16,17,18] to that of the components of molecular motors. The smallest system, the designed left-handed alpha/alpha toroid (PDB: 4YY2) [35] (99 amino acid residues), exhibited net rotation over long-time (4 μs, which amounts to 4 ms of laboratory time) microcanonical simulations carried out at a low effective temperature (100 K), keeping zero total angular momentum, akin to the cyclopropane molecule under similar conditions in an earlier work [17]. The rotation occurred about the pseudo-threefold axis of the molecule and also about the axes perpendicular to it. Changing the microcanonical regime to that of canonical Langevin simulations at T=300 K resulted in about 100 times faster changes in the total rotation angle. These results are in agreement with the experimental finding that vortex states can exist for molecules such as hydrocarbon catenanes [24] and that model protein assemblies shaped similar to molecular motors populate multiple rotational states that correspond to rotation about the system’s axle [25].

We subsequently studied protein complexes that are parts of rotating molecular motors, including the RBM2 inner ring of the flagellum MS-ring protein from *Salmonella* FliF; PDB:6SD5) [36] and the rotor of the V-Type Na${}^{+}$-ATPase from *Enterococcus hirae* (PDB: 2BL2) [12], which are composed of 22 and 10 protein monomers, respectively, arranged in symmetric rings. Both systems are located in lipid-membrane environments. The 2BL2 protein pierces the membrane and has extra-membrane sections at both membrane sides. In both systems, a protein-complex shaft is present inside the ring, which was modeled in our simulations by imposing cylindrical restraints (see Section 2.4.2), while the mean-field lipid model developed in our earlier work [37] was used to represent the lipid bilayer. Of those two systems, 2BL2 is almost a complete motor save for the stator part that relies on conformational changes due to ATP hydrolysis to the ring through protomotive forces, while 6SD5 constitutes only a small fragment of a bacterial flagellum.

The canonical simulations carried out for 6SD5 and 2BL2 demonstrated that net rotation of the ring occurs spontaneously, though its direction and extent varies depending on the initial conditions. As opposed to the 4YY2 system and to model cyclic alanine-based polypeptides studied in an earlier work [18], rotation occurred about the ring pivot in all cases. Interestingly, the pattern of rotation-angle variation did not change after replacing type II restraints with a type I restraint in which only the ring diameter, not the ring-pivot position and orientation, was controlled; in the latter case, the center of the ring moved in plane randomly. The extent of rotation was very remarkable, reaching up to $140^{\circ }$ for the 2BL2 system, clearly exceeding the standard deviations of a mean rotation angle calculated over all monomers and averaged over the respective trajectories. Interestingly, one cycle of this motor involves a rotation by $120^{\circ }$ only (with the direction of rotation depending on the phase of the cycle) [12]. These results clearly demonstrate that thermal motions can be converted into net rotation and that this rotation is organized to occur about the ring pivot, as opposed to that in the 4YY2 system studied in this work and the model cyclic alanine polypeptides studied elsewhere [18], in which cases the rotation axis changed significantly and the rotation resembled that of a sheet of paper tossed in the air during windy weather rather than that of a winged maple seed falling from a tree. Our simulations also suggest that rotation can occur in both clockwise and counter-clockwise directions, consistent with the experimentally observed behavior of bacterial flagella [66] and of bacterial ion pumps [12].

Two factors are likely to contribute to this different behavior. The 4YY2 system, although it has an oblate shape, has a diameter not much greater than its height and has little empty space near the pseudo-threefold-symmetry axis. Therefore, shape deformations due to the action of internal forces (the NVE regime) or impacts from the thermostat (the NVT regime) are likely to be converted into rotations in any direction. Conversely, each of the two motors has a large diameter, and their monomers are arranged into a relatively thin ring. Shape deformations are likely to affect the monomer interface in the first place, which is maintained by relatively weak non-covalent interactions and which is perpendicular to the ring. Thus, if net rotation emerges, it is rotation about the ring pivot. Consequently, the ring shape of molecular motors can be the key factor that enables them to convert thermal motions into net rotation. The second factor can be additional fixing of the ring plane by placing the ring in the lipid membrane. In particular, when the motor pierces the lipid membrane, substantial free-energy expense is required to distort its pivot from the perpendicular orientation to the bilayer because of the necessity of bringing some of the extra-membrane sections to the bilayer and some of the intra-membrane sections outside the bilayer.

Even though the thermal motions can be converted into net rotation, its direction and extent varies. It is likely that this is the place where the energy from ATP hydrolysis is required, which, thorough the conformational changes in the stator, controls the rotation. Further exploration of this mechanism requires considering a whole rotating motor together with the stator part. This research is planned in our laboratory.

## Figures and Tables

**Figure 1 biomolecules-13-00941-f001:**
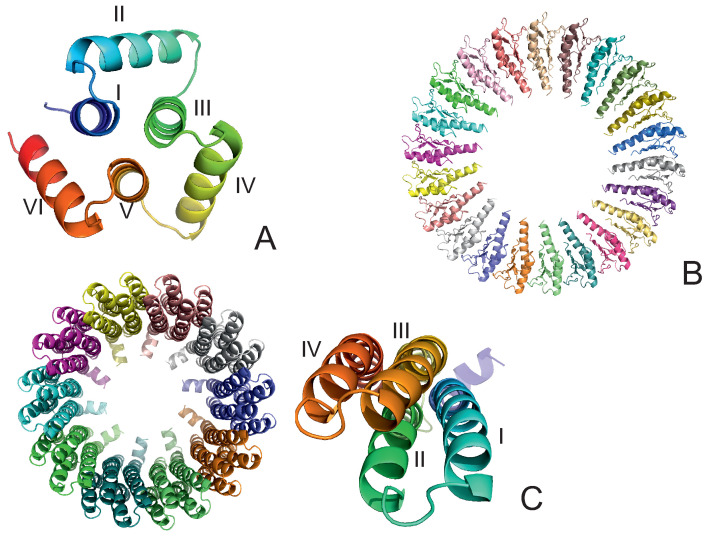
The systems considered in this study. (**A**): 4YY2, (**B**): 6SD5, (**C**): 2BL2. In panel (**A**), the helices are labeled with Roman numerals; in panel (**B**), the chains are colored differently. In the left part of panel (**C**), the chains are colored differently, and one monomer is displayed in the right part of the panel with residues colored from blue to red from the N- to the C-terminus; helices are labeled with Roman numerals.

**Figure 3 biomolecules-13-00941-f003:**
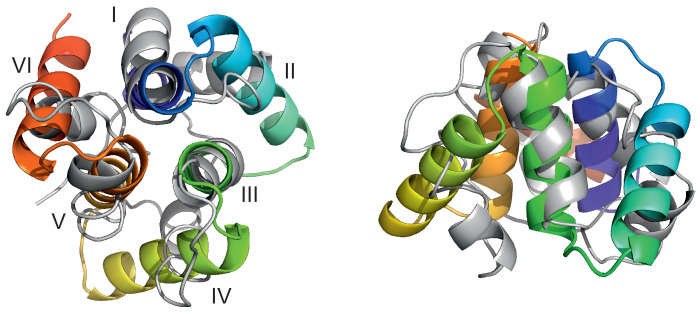
Top view (**left**) and side view (**right**) of the superposition of the predicted (gray cartoons) and experimental (cartoons colored from blue to red from the N- to the C-terminus) structures of 4YY2, chain A. The C${}^{\alpha }$-RMSD over all 99 residues present in the experimental structure is 6.7 Å. The helices are labelled with the Roman numerals in the left panel. The drawings were made with PyMOL [57] after converting the coarse-grained structures to the all-atom representation using PULCHRA [58] and SCWRL [59].

**Figure 4 biomolecules-13-00941-f004:**
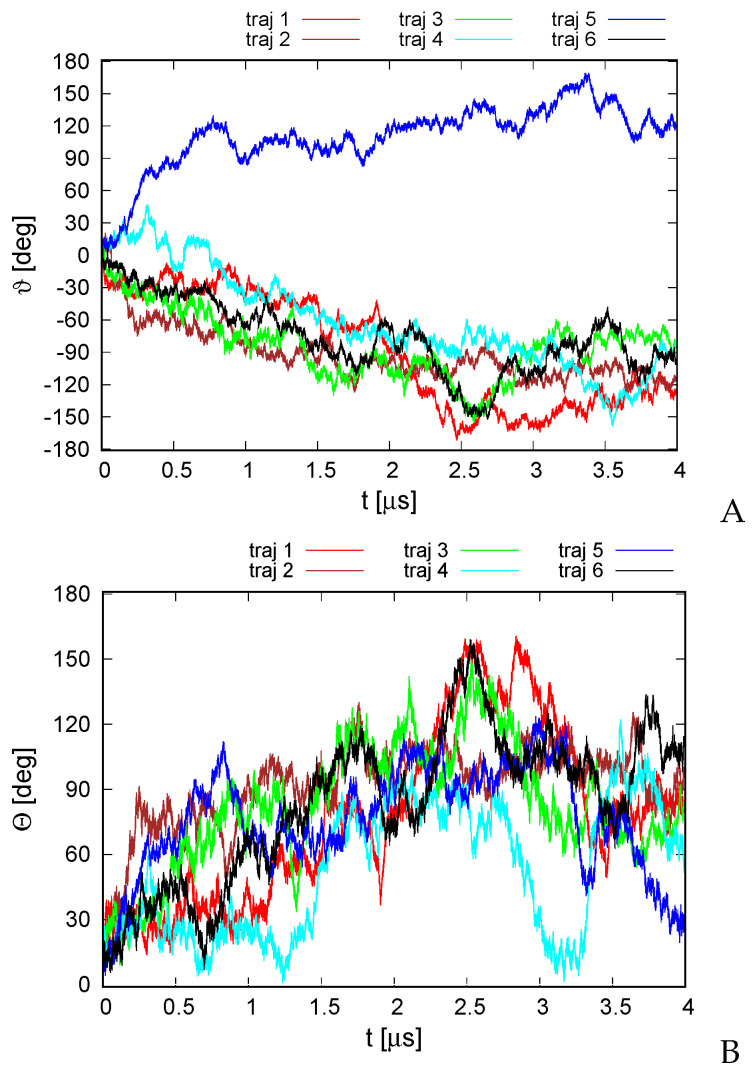
Variation of the total rotation angle ϑ (**A**) and of the Θ angle of the normal to the triangle formed by helices II, IV, and VI (Figure 1A) and the respective normal of the initial structure (**B**) with time for the six microcanonical runs of 4YY2. The drawings were made with gnuplot [65].

**Figure 6 biomolecules-13-00941-f006:**
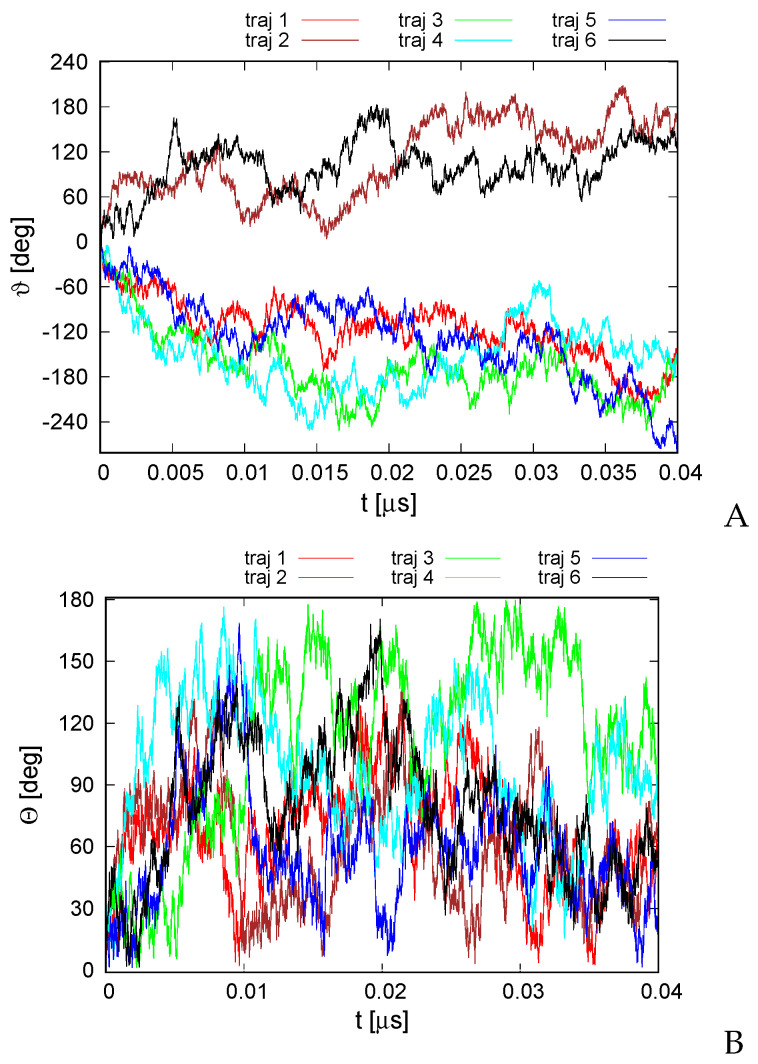
Variation of the total rotation angle ϑ (**A**) and of the Θ angle of the normal to the triangle formed by helices II, IV, and VI (Figure 1A) and the respective normal of the initial structure (**B**) with time for the six canonical runs of 4YY2. The drawings were made with gnuplot [65].

**Figure 7 biomolecules-13-00941-f007:**
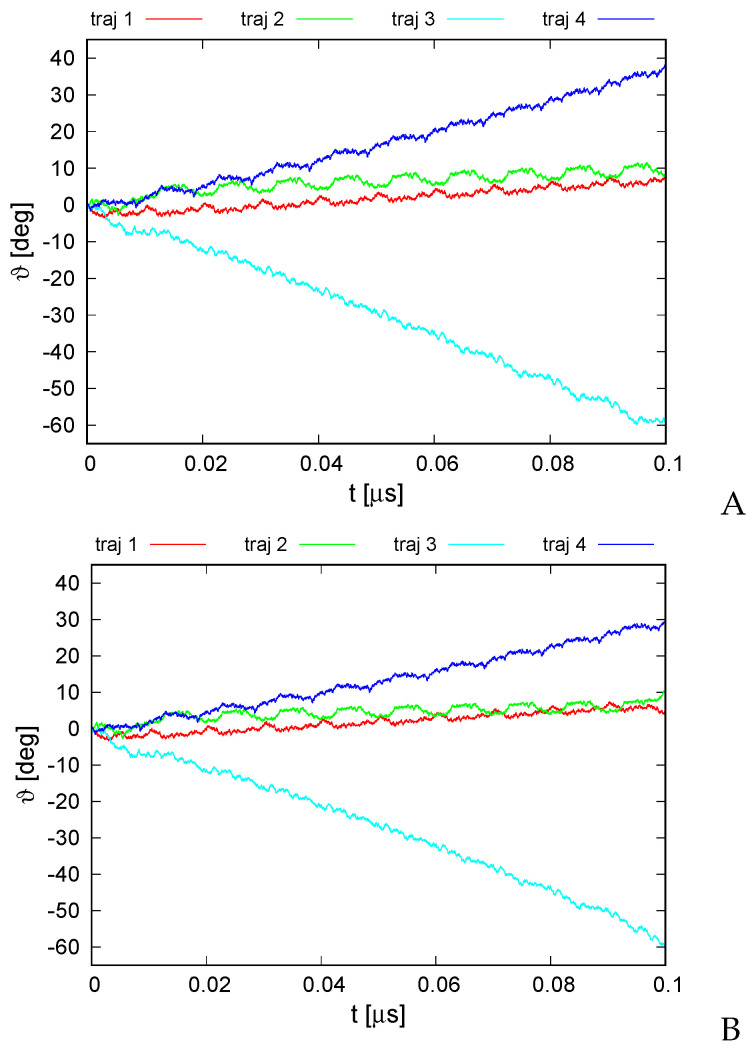
Variation of the average rotation angle ϑ of the monomers from the initial position about the ring pivot in canonical simulations of the 6SD5 system in the lipid-bilayer phase type I (**A**) and type II restraints (**B**). The drawings were made with gnuplot [65].

**Figure 8 biomolecules-13-00941-f008:**
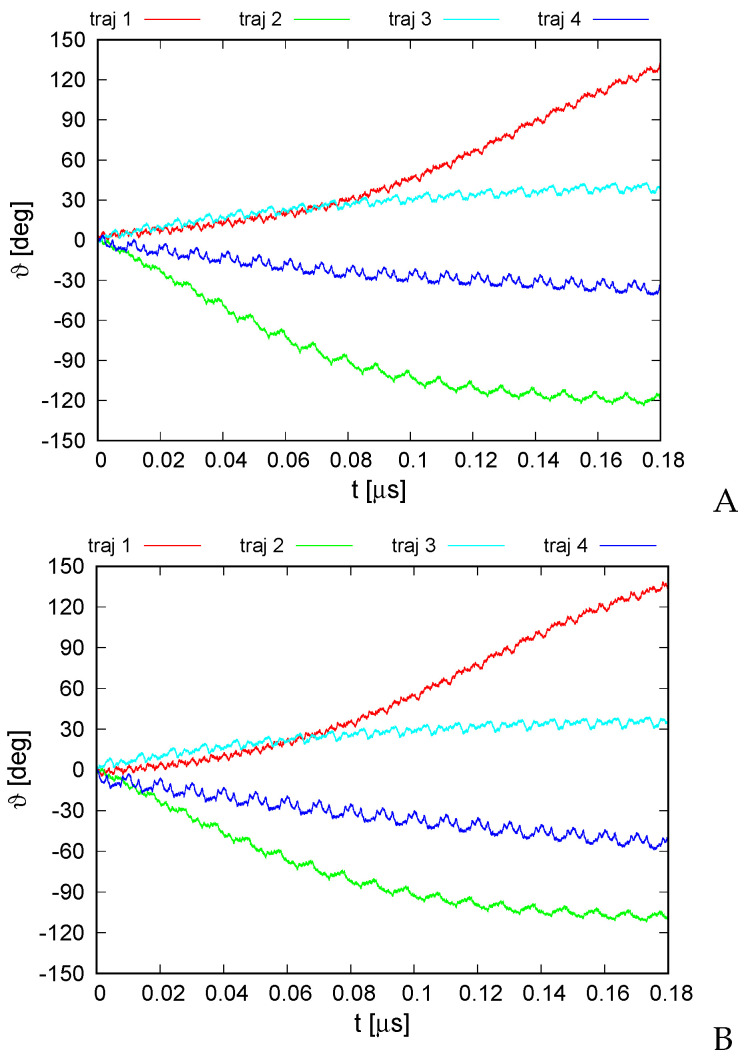
Variation of the average rotation angle ϑ of the monomers from the initial position about the ring pivot in canonical simulations of the 2BL2 system in the lipid-bilayer phase type I (**A**) and type II restraints (**B**). The drawings were made with gnuplot [65].

**Figure 9 biomolecules-13-00941-f009:**
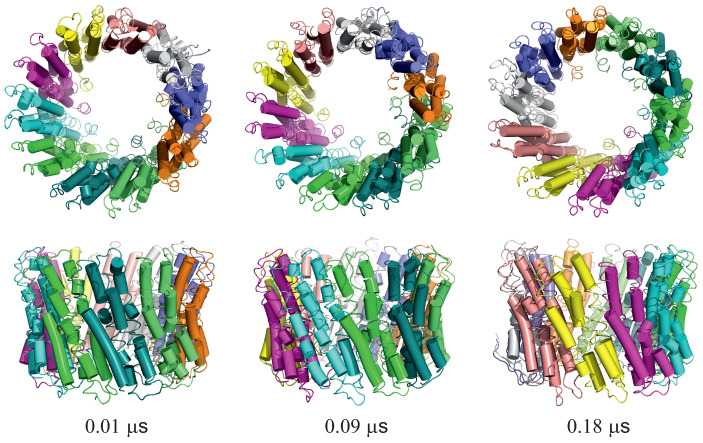
Selected snapshots from trajectory 1 of the 2BL2 system obtained in canonical simulations with the lipid-bilayer model. Type II restraints were imposed. Upper and lower panels are top and side views, respectively. Each monomer is colored differently. The drawings were made with PyMOL [57] after converting the coarse-grained structures to the all-atom representation using PULCHRA [58] and SCWRL [59].

## Data Availability

All data, including the simulated structures and trajectory analysis results, are available from the authors upon request.

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
