# Peer review of "Long-Time Dynamics of Selected Molecular-Motor Components Using a Physics-Based Coarse-Grained Approach"

_biomolecules, 2023, doi:10.3390/biom13060941_

Round 1

Reviewer 1 Report

The manuscript by Liwo et al. describes molecular dynamics simulations of three ring-like proteins acting as parts of molecular motors, including the V-type Na+-ATPase.  Due to using a coarse-grain approach, the effective simulation time approached 1 ms. The software used was previously developed by the authors and is available to users via Internet. Trajectory analysis demonstrated autorotation in the three models studied in the absence of external energy input, just under the action of thermal motions. This finding is of a paramount importance for understanding the work of these and other rotary molecular motors. The article is well thought through, written, and illustrated, and can be published as is.

Minor remarks:

The first paragraph of the section 2.6 is not quite clear without Figure 5. Some kind of mentioning in this paragraph that the pertinent illustration is found in Results may be appropriate.

Salmonella flagella (flagellum in one case) is not a Latin name of a species and should not be all italicized. The species, I presume, is Salmonella enterica. “Flagella” or “flagellar” is an adjective meaning “pertaining to flagella”.

Author Response

We would like to thank this Reviewer for his/her time and effort devoted to reviewing our manuscripts. We have implemented the suggested changes and address the comments below  Each of our reply is preceded by the respective comment in quotes. In our replies, we have explicitly indicated the page and line numbers in which the changes were made. Unless indicated otherwise, the page and line numbers refer to the revised manuscript. Additionally, we provide a version of the manuscript (in pdf format) with changes marked in red.

“The first paragraph of the section 2.6 is not quite clear without Figure 5. Some kind of mentioning in this paragraph that the pertinent illustration is found in Results may be appropriate.”

We agree with the Reviewer and in the revised manuscript we refer to the Figure 5 for the illustration of triangle and reference system construction (page 8 lines 298-300).

“Salmonella flagella (flagellum in one case) is not a Latin name of a species and should not be all italicized. The species, I presume, is Salmonella enterica. “Flagella” or “flagellar” is an adjective meaning “pertaining to flagella”.”

We agree with the Reviewer. The confusion arose because we put “flagellum” and “flagella” in italics making it a part of the species name. The correct phrase is “the RBM2 inner ring of the flagellum MS-ring protein FliF from Salmonella” and it has been implemented throughout the revised manuscript (lines 3-4 of the Abstract page 2 lines 66-67, page 3 lines 90-91 and page 15 line 497). The full species name probably is Salmonella enterica but it was not stated in the references pertaining to the 6SDS protein or in the PDB entry so we left only Salmonella.

Reviewer 2 Report

Liwo et al. performed molecular dynamics (MD) calculations in both the microcanonical (NVE) and canonical (NVT) regimes, of a 6-stranded toroidal protein (4YY2) and of two huge molecular rotors, such as that of a V-type vacuolar NA+ ATPase ( 2BL2) or the MS ring of a bacterial flagellum (6SD5). The calculations are performed using the coarse-grained UNRES force field, to simulate their dynamics in the µs(ms) time regime, without any a priori constraint on the secondary structure. According to their results, net rotational motion can occur on these protein systems, even with zero total angular momentum in the microcanonical ensemble. The notion of spontaneous rotations at equilibrium is close to that of perpetual motion so it is still debated. Some of the same authors have produced previous theoretical-computational articles stating that the Feynmann ratchet argument must be overcome by the new solid-state concept of time crystals and that of the "Guichardet connection" (that pure vibrational motions can produce specific cyclic changes in a molecule, which in turn can produce net rotations). They have previously reported data showing net rotation on small molecules and small circular peptides with pseudocylindrical symmetry, and now they extend the computation to the mentioned very large multimeric proteins in the NVE/NVT regimes.

The article is well written and methods are thoroughly explained. Potentially, this study could improve our understanding of some aspect of the most important molecular systems within the cell. However, there are a number of underlying conceptual concerns that make reliable results difficult, all the more so due to the use of relatively new MD force fields. Despite my skepticism, I have no reason to reject the paper, but on my opinion a major revision is necessary according to following points:

1. The concept regarding the existence (and persistence in time) of time crystals at equilibrium seems very difficult to extend from systems formed by coherent quantum particles, or phases of matter, to large disordered (not crystalline) multimolecular systems immersed in the crowded environment of the cell. Authors must deeply address this point, whith comments on quantum decoherence.

2. A similar related important objection concerns Guichardet's notion of connection when moving to a higher scale. Although a particular and well-defined sequence of deformations can produce net rotations even on a macroscopic scale (like the evolutions of an acrobatic diver), it is very unlikely that such precise and ordered cyclical sequences of deformations can originate, in a huge complex molecular mass, only from atomic vibrations at equilibrium

3. In this context, MD is used as such an experimental tool to validate theoretical instances. Previous studies, performed with the gromacs MD engine, with all-atom force fields on cyclic 9- and 42-alanine peptides, have indeed shown the net rotation of these peptides at equilibrium in single MD runs. Similar results follow from this paper using UNRES force fields. the RMSD of all but one trajectory is very high, which may suggest that large periodic shape changes favor net rotation. However, neither a clear rotation-related periodicity emerges from RMSD, nor has it been reported in previous works. Note that in the search for periodic changes, RMSD could be measured on subgroups of atoms or domain; if hidden by the stroboscopic effects, it would be easy to restore.  

4. The net rotation in NVE regime for 6SD5 and 2BL2 systems is negligible, but it is not negligible in NVT regime, where instead it is not possible to verify the energy conservation. As the authors know, energy drift is a common concern in MD. Can they exclude a small energy drift in the calculation, just enough to allow for rotational drift, but negligible compared to the total monitored energy? Can they provide counterexamples of molecules or proteins that don't rotate? What about proteins different from rotors, with a rough cylindrical axis such as individual alpha-helices or GPCRs? Or can't they measure any rotation about their axis, or do they expect rotation to emerge anyway according to simple rules of size and geometry?

5. The authors report that net rotations about the axis of symmetry can occur in both clockwise and counterclockwise directions. For large molecular rotors, this is in agreement with the fact that both flagella and ion pumps, products of molecular evolution, can function by rotating the rotors in either direction. In turn, the ion pump can transport ions in both directions, depending on the EC potential due to the ion concentration gradient and the internal concentration of ATP, ADP and Pi. Thus, the "rotational state" of the rotor, found in a closed system at thermodynamic equilibrium, seems irrelevant compared to the driving electrochemical forces, which regulate the ion pump in an open system away from equilibrium.

Reviewer 3 Report

The manuscript “Long-Time Dynamics of SelectedMolecular-Motor Components by Using a Physics-Based Coarse-Grained Approach” reports the results of coarse-grained simulations of motor proteins including a de novo protein. Although the topic of this paper is interesting and important in biophysics, I have concerns about the simulation procedure and discussion of the study. My main concern is about and simulation method used in 6SD5 and 2BL2 (4th comment in the following “major points”). In addition, I think that more detailed analysis and discussion are required about the molecular motions causing the rotational motion of protein. So, in my opinion, the authors should carry out an additional simulation and analysis and resubmit a manuscript.

- Major points

1.    In Section 2.4.1 (lines 194-195, page 6), the authors described "the predicted structure has the correct secondary structure," but the structures appear to be different. It would be better to show that quantitatively, for example, by showing a ratio of residues whose secondary structure corresponds to that of the reference structure using DSSP.

2.    In Section 3.1, the average RMSD for the canonical simulation was 6.8 Å, and this value is nearly the same as that for the structure prediction simulation in Section 2.4.1 (6.7 Å). If the topology of helix packing has not changed, what has changed?

3.    In the end of Section 3.1, I think that the expression “its net direction does not change” is inappropriate because the final direction of rotation depends on the region of the trajectory considered. (If we consider the first part of the simulation as a relaxation process, most of the trajectories appear to rotate randomly around a certain angle.)

4.    In Section 3.2 (lines 393-395, page 13) and Section 3.3 (lines 421-423, page 14), the authors described “these patterns seem to result from the fact that the random-number generator was restarted with every run restart, which occurred after every 10,000,000 MD steps.”

If the trajectories are periodic because the same numbers are generated periodically, then the rotation in one direction could be an artifact of simulation. I think that the simulation should be redone and verified,
for example, by using the system clock as seed values for a random number generator. (Why wasn’t the same phenomenon observed in the 4YY2 system?)

5.    What do “corresponding trajectories” in Line 385 of page 13 and “their counterparts” in Line 419 of page 14 mean? Do trajectories with the same index have the same initial velocity?

6.    The authors described that the shape deformation leads to the rotational motion, but the RMSD values were quite small in the trajectories with type II restraint of the 6SD5 system. To what extent did each protomer undergo a conformational change?

As described above,
I think that more detailed analysis and discussion are required about the molecular motions causing the rotational motion of protein.

7.    To indicate that the property observed in this study is a unique to motor proteins, it is required to be verified that no other rotationally symmetric oligomeric protein exhibits this behavior. If there are previous studies, they should be mentioned.

8.    From the result of 6SD5 it appears that the motor keeps rotating in one direction only with thermal energy, and the authors concluded that the energy from ATP hydrolysis is required only to set the direction and extent of rotation. Does this mean that the bacteria would be able to move only with thermal energy?

9.    I think the energy source of flagellum movement is not an ATP hydrolysis but a protonmotive force.

- Minor points

1.    The lengths of the simulations differ among systems. How were they determined? (Why was the duration of the canonical simulations only 0.04 µs, even though the simulation was run for 0.1 or 0.18 µs on larger systems?)

2.    In the caption of Figure 1, the presence or absence of parentheses is not consistent for A, B, and C.

3.    In Section 3.1, I think that it is better to add an explanation for the structural transition observed around 0.3 µs in the trajectory 3.

4.    In Line 323 in page 10, "are" is duplicated. (For illustration, selected structures from trajectory 4 are are...)

5.    I think it is better that Equation 5 is integrated and simplified (ij=-ji=k, jk=-kj=i, ki=-ik=j).

6.    Equation 7 should be corrected.

7.    An explanation is required for atan2 in Equation 8.

8.    Lines 367-368 in page 12, it is better to add the average RMSD value of the protomers.

9.    Lines 369-370 in page 12, the pairs of parentheses do not match.

10. Line 418 in page 14, there is a typo (SD5 -> 6SD5).

11. The title of ref. 18 is probably wrong. (From Feynman's ratchet to timecrystalline molecular motors)

Reviewer 4 Report

The manuscript is well written and reports extensive evidence that thermal motions can be converted into net rotation for protein components of molecular motors. Although the UNRES model does not take into account the specificity of molecular driving forces, and is, by definition, an implicit solvent model, the authors make an effort to clarify that these important features are not strongly influencing the properties they refer to. 
I believe that this is the main merit of the present manuscript: they are not trying to oversell the UNRES model, but rather employ it to overcome limits of the more rigorous all-atoms approach that would not yield the same time intervals explored in the present study.
I therefore recommend this manuscript for publication.

Author Response

We would like to thank this Reviewer for his/her time devoted to assess our manuscript and for his/her favorable decision. As far as we can see, the Reviewer did not request any revision. The Reviewer was right in his/her conclusion that we employed the coarse-grained UNRES force field to be able to study long-time behavior of the molecular motors, as is stated in the Introduction section of the manuscript.

Reviewer 5 Report

The manuscript presents a computational study of the movement of 3 selected molecular-motor components. The authors have approached this problem using long-scale coarse-grained molecular dynamics simulations.

I found the manuscript extremely interesting and very well put together. The research topic is really fascinating and I personally, as a computational biophysicist, find that the presented by the authors results are very intriguing. The paper is organized very well, having a logical and coherent presentation with excellent and informative analyses and graphics. I was not able to notice any issues with the manuscript.

Therefore, I highly recommend publishing the manuscript "Long-Time Dynamics of Selected Molecular-Motor Components by Using a Physics-Based Coarse-Grained Approach" in the Biomolecules journal as submitted.

Author Response

We would like to thank this Reviewer for his/her time and effort devoted to assess our manuscript and for his/her favorable assessment. As far as we can see, the Reviewer did not request any revision.

Round 2

Reviewer 2 Report

my concerns have been addressed from authors

Reviewer 3 Report

Thank you for your response.